# Ultrasound and Bioptic Investigation of Patients with Primary Sjögren’s Syndrome

**DOI:** 10.3390/jcm10061171

**Published:** 2021-03-11

**Authors:** Valeria Manfrè, Ivan Giovannini, Sara Zandonella Callegher, Michele Lorenzon, Enrico Pegolo, Alessandro Tel, Saviana Gandolfo, Luca Quartuccio, Salvatore De Vita, Alen Zabotti

**Affiliations:** 1Rheumatology Clinic, Department of Medicine, University of Udine, c/o Azienda Sanitaria Universitaria Friuli Centrale, 33100 Udine, Italy; manfre.valeria.31@gmail.com (V.M.); giovannini.ivan@spes.uniud.it (I.G.); sarazandonella@gmail.com (S.Z.C.); saviana.gandolfo@asufc.sanita.fvg.it (S.G.); luca.quartuccio@asufc.sanita.fvg.it (L.Q.); zabottialen@gmail.com (A.Z.); 2Institute of Radiology, ASUFC Udine, 33100 Udine, Italy; michele.lorenzon@asufc.sanita.fvg.it; 3Institute of Anatomic Pathology, ASUFC Udine, 33100 Udine, Italy; enrico.pegolo@asufc.sanita.fvg.it; 4Maxillofacial Surgery, Department of Medicine, University of Udine, c/o Azienda Sanitaria Universitaria Friuli Centrale, 33100 Udine, Italy; alessandro.tel@icloud.com

**Keywords:** Sjögren’s syndrome, salivary gland, salivary swelling, histopathology, ultrasonography, biopsy, US-guided core needle biopsy, lymphoma

## Abstract

Primary Sjögren’s syndrome (pSS) is a chronic and heterogeneous disorder characterized by a wide spectrum of glandular and extra-glandular features. The hallmark of pSS is considered to be the immune-mediated involvement of the exocrine glands and B-cell hyperactivation. This leads pSS patients to an increased risk of developing lymphoproliferative diseases, and persistent (>2 months) major salivary gland enlargement is a well-known clinical sign of possible involvement by B cell lymphoma. Better stratification of the patients may improve understanding of the mechanism underlying the risk of lymphoproliferative disorder. Here, we summarize the role of different imaging techniques and a bioptic approach in pSS patients, focusing mainly on the role of salivary gland ultrasonography (SGUS) and a US-guided core needle biopsy (Us-guided CNB) as diagnostic and prognostic tools in pSS patients with persistent parotid swelling.

## 1. Introduction

Lymphoma Risk in pSS Patients, and the Role of Persistent Salivary Gland Swelling

Primary Sjögren’s syndrome (pSS) is a chronic systemic autoimmune disease, affecting mainly middle-aged perimenopausal women with a female to male ratio of 9:1, and has an estimated prevalence of about 0.5% in the general population [1].

The clinical features can be heterogeneous, ranging from glandular (mucosal dryness, glandular swelling) to systemic multi-organ manifestations (such as musculoskeletal, renal, neurological and pulmonary involvement, and cryoglobulinemic vasculitis), and even lymphoproliferative disorders [2]. The hallmarks of pSS pathogenesis are the immune-mediated destruction of exocrine glands and B-cell hyperactivation, leading pSS patients to have an increased relative risk (7–19 fold) [3] of developing non-Hodgkin’s lymphoma (NHL). The lifetime risk has been estimated between 5 and 10% in various studies, representing the highest susceptibility among autoimmune disorders [4]. The most represented NHL histotype in pSS is an indolent, low-grade extra-nodal marginal zone (MZ) B-cell lymphoma of mucosa-associated lymphoid tissue (MALT) that is mainly localized in major salivary glands (SGs), and the parotid in particular, or, less frequently, in lacrimal glands, lungs and stomach; some could transform into high-grade lymphoma, mainly diffuse large B-cell lymphoma, which may also originate from MZ NHL [5,6,7,8,9].

Salivary gland lymphoma usually presents itself as a unilateral asymptomatic mass or bilateral persistent gland swelling [10]. Lymphoma development appears as the final stochastic result of a chronic inflammation in the SGs, which is facilitated by an adverse, aggressive profile that may present early in the pSS disease course with specific clinical, biological and genetic features of stronger B cell hyperactivity [11].

Among the clinical manifestations, persistent SGs swelling, especially parotid enlargement, is considered a major risk factor for the development of MALT lymphoma [12], as highlighted by Kassan SS et al. in 1978 [1]. The other main feature predicting NHL in pSS is mixed cryoglobulinemia, particularly with associated vasculitis. Rheumatoid factor (RF) positivity and C4 hypocomplementemia may also reflect the expansion of peculiar B cells that are more prone to transformation. Other highlighted risk factors are splenomegaly, lymphadenopathy, leukopenia, anti-Ro/SSA or/and anti-La/SSB positivity, monoclonal gammopathy, increased B2-microglobulin levels and increased free light chain k/l ratio [1,13,14,15]. Independent risk factors for lymphoproliferation are also a high inflammatory infiltration within the glandular parenchyma, represented by a high focus score (e.g., Focus Score ≥ 3) [16,17], as reported in Table 1.

Many recent studies have focused on the elaboration of novel methodologies and algorithms to predict and stratify lymphoma risk in pSS patients (https://www.harmonicss.eu accessed on 13 January 2021) [8,9,13,18,19]. 

The use of the ESSDAI appears to reflect the aforementioned lymphoproliferative disease and seems to be more an accompanying factor of lymphoma development rather than a true predictor [15,20]. Furthermore, baseline ESSDAI results are low in about 30% of pSS who develop NHL, while persistent SGs swelling and cryoglobulinemia appear to be more relevant [21]. In turn, cryoglobulinemia in pSS appears to depend on MALT, and, in this particular characteristic, differs from the classical cryoglobulinemia related to HCV infection. The thorough study of the SGs and MALT is of primary importance for the issue of lymphoma prediction in pSS.

To this end, in this paper, we will review current innovations in SG imaging and bioptic techniques for pSS investigation, aiming to achieve an improved disease stratification and prediction of lymphoma development. We focused mainly on novel innovations in SG ultrasonography (US) and US-guided core needle biopsy (Us-guided CNB) in pSS. Patients with persistent SG swelling may particularly benefit from these innovations. The research was concentrated on the literature published in the last five years (research conducted on the PubMed database on the following terms: Sjögren’s syndrome; salivary gland; salivary swelling; histopathology; ultrasonography; biopsy; US-guided core needle biopsy; lymphoma).

## 2. Imaging and Parotid Swelling

### 2.1. Imaging and Parotid Swelling a Cross-Sectional Relationship

It has recently been shown that imaging techniques can assist in the diagnosis of pSS, and in the assessment of the activity and progression of the disease. Furthermore, many imaging techniques have been evaluated as fundamental tools in diagnosis and staging of pSS-associated SGs lymphoma [22]. 

When encountering a patient affected by parotid gland swelling and pSS, a thorough clinical evaluation is the first approach, followed by a radiological and, if needed, a pathological work-up. In this chapter, the Authors focused on ultrasonography (US) as one of the most relevant imaging techniques for the examination of parotid swelling and of its most dreadful consequence, i.e., the development of SGs lymphoma, in pSS, followed by Magnetic Resonance Imaging (MRI). A brief insight into the actual role of sialography and nuclear medicine in pSS is also given.

### 2.2. The Relevance of Salivary Gland UltraSound (SGUS) in pSS

The usefulness of SGUS in pSS was highlighted by us almost 30 years ago [23], but in the last 10 years, salivary gland ultrasound (SGUS) has definitely proved effective for the evaluation of structural abnormalities and for the detection of SGs involvement in patients with suspected or established pSS. SGUS is a simple, non-invasive, non-irradiating, inexpensive and repeatable technique [24,25,26,27,28], and various studies have demonstrated that its inclusion in pSS ACR-EULAR classification criteria improves performance, diagnostic accuracy and feasibility, increasing sensitivity from 87.4% to 91.1%, when physician diagnosis was the reference standard [26,29,30,31].

However, SGUS is not yet part of the classification criteria for pSS, mainly due to the sizable intra- and inter-rater disagreement among less-expert sonographers, and particularly, in some sonographic features such as the posterior border evaluation and the grading of hyperechoic bands [28,32,33].

Recently, the OMERACT SGUS task force group filled the gap: a new four-grade semiquantitative score was proposed with good/excellent agreement results (Light’s kappa 0.81; 95%CI 0.77 to 0.84 for intra-reader reliability; Light’s kappa 0.66; 95%CI 0.61 to 0.70 for inter-reader reliability) [34].

The same score, as well as the earliest and simplest SGUS score of 1992 by De Vita et al. were recently tested and compared in a reliability exercise sponsored by HarmonicSS [23,35]. Twenty-seven European sonographers participated in the study and the results confirmed that the two SGUS scores are a reliable tool. Furthermore, in this study, the agreement was independent from the years of experience of the sonographers and of their previous use of the tested scores. An electronic manual given preliminarily to the participants was sufficient for training and reference, if needed. Currently, the SGUS scores are set and tested for diagnostic purposes, and only few data are published on their roles for follow-up, prediction of treatment response or as surrogate of glandular damage detected by histopathology [24].

Regarding this last item, SGUS scores and damage-related lesions have been reported as being correlated with objective SGs function (i.e., reduced salivary flow rate) [36]. An intriguing role of SGUS in pSS is its possibility to repeatedly and non-invasively assess glandular involvement, differentiating between activity and damage, and thus better stratifying patients during the follow-up [37,38,39].

Many authors have analyzed the connection between SGUS features and patient’s serology (presence of ANA, anti-Ro/SSA and/or anti-La/SSB antibody, RF, CD4+ lymphopenia and reduced number of memory B cells, thrombocytopenia and C4 complements levels), clinical features (lower salivary flow rate, SG swelling, skin vasculitis, an ESSDAI value > 5), and histological findings (presence of GC-like structures in MSG biopsy), many of them being well known lymphoma risk markers, as previously described [24,40,41,42,43].

In the authors’ opinion, SGUS is also crucial to assist clinicians in the detection of glandular lymphoma, although few studies on this have been published to date. Lymphoma detection by SGUS in pSS has been described in some case series: they individuate, as expected, mostly MALT lymphoma (with two typical patterns described: the “linear echogenic strands pattern” also referred to as “multiple small hypoechoic nodules” or “tortoiseshell pattern”, and the “segmental pattern”/“multiple layer hypoechoic masses”) although no generally defined sonographic criteria for parotid lymphomas exist, as they present various non-specific US appearances [10,44,45,46]; see Figure 1.

However, in a recently published series, MALT lymphoma detected by SGUS also appeared as focal, hypoechoic, and dishomogeneous, with hypervascular areas with posterior acoustic enhancement, variable morphology and margins, and internal septa in half of the cases [47]. Importantly, only specific areas, and not all of the parotid gland, appeared to be involved by SGUS, which led to a new application of SGUS in pSS: to guide major SGs biopsy in definite glandular areas, usually in pSS patients with persistent parotid or submandibular enlargement.

There is a current, great effort to achieve automatic scoring of SGUS in pSS. Artificial intelligence is a promising tool for an improved evaluation of SGUS in pSS. The HarmonicSS project is developing image segmentation to this end, i.e., automatic, computer-assisted evaluation and scoring of SGUS images after machine learning. This will possibly help to overcome the still open problem of operator reliability in SGUS. Published initial studies report very encouraging data to this end [24,48,49].

### 2.3. Magnetic Resonance Imaging

Magnetic resonance imaging (MRI) has been used since 1996 to characterize SGs in pSS and can be useful both in early and late stages. Although informative in pSS, MRI is not routinely performed since it is expansive and of difficult accessibility. Considering its ability to explore the deeper region of the SGs and its panoramic view, it can have a role in the evaluation of pSS-related lymphomas of the head and neck. MALT lymphomas of the parotid gland are characterized by variations of contours and internal structures of the masses on MRI: solitary solid mass, multiple solid nodule or masses, solitary solid-cystic mass and diffusely solid-cystic lesion, with the latter two being more common. Non-MALT parotid lymphomas are identified mainly as solitary, well defined masses of uniform density; sometimes they show necrotic areas within the matrix, and are usually accompanied by enlarged and fused cervical lymph nodes [50,51,52].

In MRI, parotid lymphomas have homogeneous intermediate-signal intensity and an enhancing rim on the post contrast T1-weighted images and low signal intensity on the T2-weighted images without obvious enhancement effects [52,53]. Multimodal MRI examination (functional MRI, dynamic contrast-enhanced MRI, diffusion-weighted imaging, and apparent diffusion coefficient calculation) can increase differentiation accuracy of parotid lesions [53,54,55]. Although benign and malignant lesions of the SGs often appear similar, with overlapping and non-specific features in MRI, the high spatial resolution and fine characterization of deeper anatomical structures make MRI an imaging technique that can be useful in some cases for diagnostic guidance and local staging of pSS-associated SGs lymphoma [22,56]. Recently, as SGs features in MRI showed concordance with salivary glands ultrasound (SGUS), SGUS might represent a more accessible alternative for pSS assessment both in the diagnostic iter [22] and in guiding the histological sampling of suspicious lesions.

### 2.4. Sialography and Nuclear Medicine Techniques

Traditionally, sialography and scintigraphy represented the main imaging techniques to evaluate salivary glands, as included in the 2002 AECG criteria. More recently, the ACR/EULAR classification criteria excluded their role due to their multiple drawbacks.

Sialography is, in fact, characterized by invasiveness, high risk of complication and radiation, and multiple contraindications [25,57]. Its role has been taken by alternative sialographic techniques as sialo-cone-beam computerized tomography and magnetic resonance sialography: both are less invasive and have a higher spatial resolution, providing three-dimensional images [58,59,60].

Scintigraphy was omitted by the 2016 ACR/EULAR criteria due to its low specificity and inability to differentiate uptake failure from secretory failure [61] Several studies analyzed its usefulness in the determination of salivary gland disfunction and its correlation with pSS severity [62,63,64]; the role of specific scintigraphic tracers has also been evaluated, as 99mTc-HYNIC-TOC, which binds to the overexpressed somatostatin receptors [65]; 99mTc labeled with Rituximab, which images B lymphocytes [66]. Nevertheless, the limited resolution (8–10 mm) and the inability to quantify the exact uptake remain major issues [22,56].

Several authors reported the role of FDG-PET/CT in detecting disease activity in pSS, in screening pSS patients for lymphoma, guiding biopsy, in system staging of pSS-associated lymphoma and monitoring response to treatment. Recently, Keraen et al., in a retrospective study on pSS patients, showed that a SUVmax ≥4.7 in the parotid glands and presence of focal lung lesions were associated with lymphoma [67]. As for scintigraphy, pSS-specific PET tracers with the capacity to bind B and T cells (e.g., 89Zr-Rituximab [68,69], radiolabeled IL-12 [70]) have been evaluated, and further researchers in the field could allow promising insights on the pathogenesis of the disease [71].

## 3. Role of Ultrasound-Guided Core Needle Biopsy of Salivary Glands in pSS

SGUS provides the preliminary identification of suspect SG lesions of various nature in pSS, assessing in detail their features and subsequently guiding tissue sampling. Fine-needle aspiration cytology (FNAC) of the parotid gland is an established technique that provides material for cytological analysis, but current evidence shows that histological assessment has superior diagnostic potential [25,72,73,74]. Thus, when more conspicuous tissue sampling is required, the clinician has to resort to different procedures [75].

Open surgical biopsy is still not routinely performed, both due to the lack of expertise and to the possible complications, mainly injury of the facial nerve. Moreover, the collected glandular parenchyma may not include the damaged glandular area(s).

Ultrasound-guided core-needle biopsy (US-guided CNB) represents an effective, established and safe alternative to open surgical biopsy. It preserves tissue architecture, allowing for immunohistochemical staining to be carried out. Considering pSS as a lymphoproliferative disorder, the histologic examination is useful not only for diagnostic purposes but also for the staging and grading of neoplasms; the amount of harvested tissue also permits flow cytometry analysis, which is fundamental in the evaluation of lymphomatous lesions [76]. In fact, this technique is currently performed in the diagnostic work-up of epithelial tumors of major SGs with an estimated sensitivity and specificity of 96% and 100%, respectively, accordingly to Witt et al.’s meta-analysis [77].

In a meta-analysis of Kim et al., 1315 patients were subjected to US-guided CNB of SGs, of which 83% were of the parotid gland; only one case of temporary facial weakness due to local anesthesia of the facial nerve was reported, and only seven cases of hematoma. According to these results, US-guided CNB might be better tolerated in comparison to surgical biopsy [78], so the role and feasibility of US-guided CNB as a diagnostic and prognostic tool in pSS have been studied. Baer et al. [76], in a retrospective study, evaluated the safety and utility of US-guided CNB for diagnosis of SG lymphoma in pSS. They performed US-guided CNB on 24 patients with diffuse persistent parotid gland enlargement or a discrete parotid or submandibular lesion identified clinically or by US. Tissue specimens were subjected to standard surgical pathology (histopathology, immunohistochemistry), molecular analysis and flow cytometry. The results demonstrated that US-guided CNB, inclusive of flow cytometry, is a useful diagnostic tool, providing sufficient material to discern a range of SG pathologic findings in pSS, and allowing a definitive pathological diagnosis and thus a clinical decision making without further testing. It permitted the differentiation between a benign SG process from a MALT lymphoma or a possible lymphoproliferative disorder in the majority of the patients. The ultrasonographic guide enabled targeting of the most suspicious area of the gland, even when not clinically evident. The Authors also reported that none of the patients reported complications from the procedure.

Recently, Zabotti et al., evaluated the diagnostic accuracy and safety of US-guided CNB in pSS patients, comparing the technique with open surgical biopsy [47]. The Authors selected a prospective case cohort of thirteen patients with pSS and clinically persistent (≥2 months) SG enlargement (nine parotid and three submandibular), who underwent US-guided CNB, confronting data with a retrospective control cohort of 13 patients who underwent parotidean open surgical biopsy. Importantly, it was specified that at least one of the US-guided CNBs was targeted at the most sonographically suspicious lesion, when present, or on the clinically most swollen gland. In all cases, the tissue samples were sufficient and adequate, permitting histopathological diagnosis, and immunohistochemical and molecular analysis. These evaluations allowed for a definite diagnosis in all the cases, and the complete characterization of a possible lymphoproliferative disease (Figure 2).

Benign and malignant lesions were easily discerned, and a rare entity, such as IgG4-related-disease, was detected. The US guide permitted the evaluation of specific and multiple targets, extending the site of sampling to unexplored areas, thus reducing the possibility of false-negative results. The five cases diagnosed as MALT lymphomas by histopathology showed features of suspected lymphoma on US evaluation (hypoechoic nodule with hypertrophic lymphoid structure, or a limited area with large and confluent hypoechoic lesions), similar to those described by Baer et al. [76]. Of note, MALT lymphoma was diagnosed only in the biopsy taken on the most suspicious focal lesion and not in the surrounding glandular parenchyma in one patient. This finding highlights the importance of a US-guided approach in performing a major SGs biopsy and that specific sonographic features might correlate with equally specific pathologic patterns. In the cohort of 13 controls who underwent open surgical biopsy in the same study, the procedure provided sufficient tissue for pathological evaluation in 12/13 patients, but was taken from a restricted glandular region (“safety zone”), which cannot effectively represent the whole disease affecting the gland.

Finally, the US-guided CNB was described as well tolerated, and it was also performed in the submandibular glands, when swollen. Only transient complications were noticed in six cases (i.e., local transient swelling and mild local transient pain), and no serious and/or persistent complications were reported. Conversely, the open surgical biopsy of the parotid gland in the safe area caused persistent sensitive complications in 2 controls and transient complications in 12 of them.

## 4. The Importance of Parotid Biopsy in pSS

SGs biopsy is fundamental for the classification and diagnosis of pSS, as it plays a prominent role in the AECG and current ACR-EULAR classification criteria, especially in seronegative patients; its prognostic value and its use in monitoring treatment response have also been intensely evaluated [61,79,80,81].

Traditionally, as firstly introduced by Chisolm and Mason [82] in the pSS diagnostic work-up, a minor salivary gland (MSG) biopsy is performed, under local anesthesia, through a small incision of the mucosa of the inner lower lip. Various surgical techniques have then been described, with the approach of Greenspan et al. [83] and Daniels [84] being the most widely applied, and no comparative studies have supported the advantages of one technique over the other [85]. Focal lymphocytic sialadenitis (FLS) represents the histological hallmark of pSS, and it is characterized by the presence of one or more dense aggregates of ≥50 lymphocytes (focus) in the histological section, usually located in the perivascular or periductal areas, while the surrounding tissue is composed by unaffected parenchyma; a Focus Score (FS) can be calculated considering the total number of foci per 4 mm^2^ of salivary gland tissue [82,83].

According to the 2016 ACR-EULAR criteria, MSG biopsy is defined positive if a FS ≥ 1 is established. In 2017, Fisher et al. in the Sjögren’s histopathology work-shop group from the EULAR Sjögren’s syndrome study group (eSSential), published a consensus guidance to support a standardization of the procedure and of the histological assessment of MSG biopsy. These guidelines recommend the harvest of a minimum of four MSGs and the analysis of a minimum 8 mm^2^ surface area of gland section; multiple cutting levels are also supported in case of an inconclusive biopsy; the overall area of glandular inflammation should be evaluated, and the presence and extent of atrophy, fibrosis, duct dilatation, non-specific chronic sialadenitis, germinal center (GC)-like structures and lymphoepithelial lesions (LELs) should be reported [86].

According to the systematic review of Guellec et al., MSG biopsy sensitivity ranges from 63.5% to 93.7%, while specificity varies from 61.2% to 100%; positive predictive value and negative predictive value range, respectively, from 74.2% to 100% and from 39.1% to 96.1%. Ref. [87] Costa et al. showed good reliability of MSG biopsy (intra-observer k values 1 for FS and 0.80 for Chisholm and Mason’s grading system; inter-observer k values of 0.71 for FS and 0.64 for Chisolm and Mason’s grading system) [88]. Moreover, the prognostic value of MSG biopsy has been thoroughly analyzed: higher FS values correlate with specific extra-glandular (Raynaud’s phenomenon, vasculitis, lymph node or spleen enlargement) and laboratory (leukopenia, anti-SSA/SSB and RF positivity) features and with the risk of NHL development [16,81,89]. The role of GC-like structures and LELs in MSG biopsy in lymphoma prediction in pSS was highlighted, but additional study is needed because of conflicting data [81,89,90].

A major limitation of MSG biopsy is, in our opinion, that random glands are investigated. In other words, minor salivary glands are randomly sampled in the biopsy procedure and the samples might not well reflect SG activity, damage and lymphoma risk. Moreover, the samples include numerous minor salivary glands, that may present differences in disease involvement. So, the analysis of minor salivary glands with various characteristics from the same patient might not be homogeneously studied and compared according to different approaches (e.g., the study of histopathology vs. gene expression). Finally, MSG biopsy is not devoid of complications, the most common being sensory alteration of the lower lip, local pain, swelling, hematoma, formation of granulomas and internal scarring [86,91]. The localized sensory alteration can be permanent in up to 10% of cases [92].

At present, biopsy of the major SGs in pSS (parotid glands being the more relevant) is not part of the current classification criteria for pSS, since it is mostly reserved for NHL investigation in cases of persistent glandular enlargement in pSS or with SG-suspected lesions. The parotid biopsy approach is not commonplace because of the risk of facial nerve damage, development of sialoceles, salivary fistulae, and temporary sensory alteration in the incision area [93], but in expert hands and with particular approaches in the “safe area”, it seems to be much less risky. In addition, the literature focuses on surgical parotid gland biopsy, while little information is found about submandibular or sublingual gland biopsy [91,92,94].

In 2006, Pijpe at al. evaluated the diagnostic value and morbidity of parotid biopsy compared with labial SG biopsy in pSS patients. The parotid gland biopsy was considered positive for pSS diagnosis if FS ≥1 or small lymphocytic infiltrates, not fulfilling the criterion of a FS ≥1, in combination with the presence of benign LELs are found in the histological specimen. The application of these histological criteria showed a sensitivity of 93%, specificity of 95%, PPV and NPV values of 93% and 95%, respectively, making it comparable to MSG biopsy [93].

More recently, van Nimwegen et al. confirmed good values of sensitivity and sensibility when MSG and parotid biopsies were performed, mostly simultaneously, to fulfil 2016 ACR-EULAR criteria. Using MSG biopsies, ACR-EULAR showed 87% absolute agreement (k = 0.73) with expert classification, with a sensitivity of 97% and specificity of 83%. When parotid biopsies were considered, the ACR-EULAR score was found to have excellent accuracy, with an AUC of 0.97 (95% CI: 0.92, 1.00) to discriminate pSS from non-pSS, and an absolute agreement of 92% (k = 0.82), 91% sensitivity and 92% specificity. As these data show, the specificity is higher when using parotid gland biopsies [95].

Compared to MSG biopsy, parotid gland biopsy allows better detections of LELs and GCs-like structures [92,93,96,97], whose role in lymphomagenesis is under strict surveillance. Consequently, since lymphoma in pSS commonly affects the parotid gland, a parotid biopsy might be preferable to MSG biopsy in patients at high risk of lymphoma. It is also repeatable, permitting the comparation of more histological findings on the same gland, and the evaluation of the response to treatment, in contrast to a biopsy of labial glands [98,99]. Moreover, it allows a direct comparison to be made with other diagnostic and functional features derived from the same gland (e.g., secretory function, sialography, scintigraphy, US, MRI) [92].

US-guided CNB of the parotid glands may represent a further advance or alternative with respect to a surgical parotid biopsy, being focused (i.e., investigating glandular areas which appear different by SGUS) and seemingly less invasive. In addition, the same approach is likely useful not only for the thorough investigation of lymphoma in pSS, but also for improved diagnosis, follow up and response to therapy in clinical trials. Further studies are therefore upcoming.

## 5. Conclusions and Further Researchers

In pSS, imaging, and SGUS in particular, has demonstrated its usefulness over the years in the diagnostic and classification process, and now it needs a formal placing. The feasibility and non-invasiveness showed by ultrasound may be relevant for monitoring pSS patients, and in assessing activity, damage and lymphoma risk. Patients’ stratification would be greatly improved. Secondly, the automatic scoring of SGUS, by means of artificial intelligence, is being actively investigated and could be achieved, allowing possible problems linked to SGUS reliability to be solved. Thirdly, based on data provided by surgical parotid biopsy in pSS, such biopsy appears relevant for a thorough evaluation of pSS disease. Since SGUS is of major value in guiding interventional bioptic procedures, the US-guided CNB of parotid/submandibular SGs represents a novel approach to pSS. By two preliminary studies, this technique proved to be an accurate, patient-friendly and safe diagnostic tool in pSS patients with persistent salivary gland enlargement. The positive results in terms of safety and feasibility, as well as the possibility to better characterize the area of tissue sampling, also support further research on the application of this technique for the whole management and stratification of pSS, besides lymphoma assessment. Strikingly, improvement for the more focused collection of SG tissue samples could be also achieved, and this is of major relevance for future biologic studies.

## Figures and Tables

**Figure 1 jcm-10-01171-f001:**
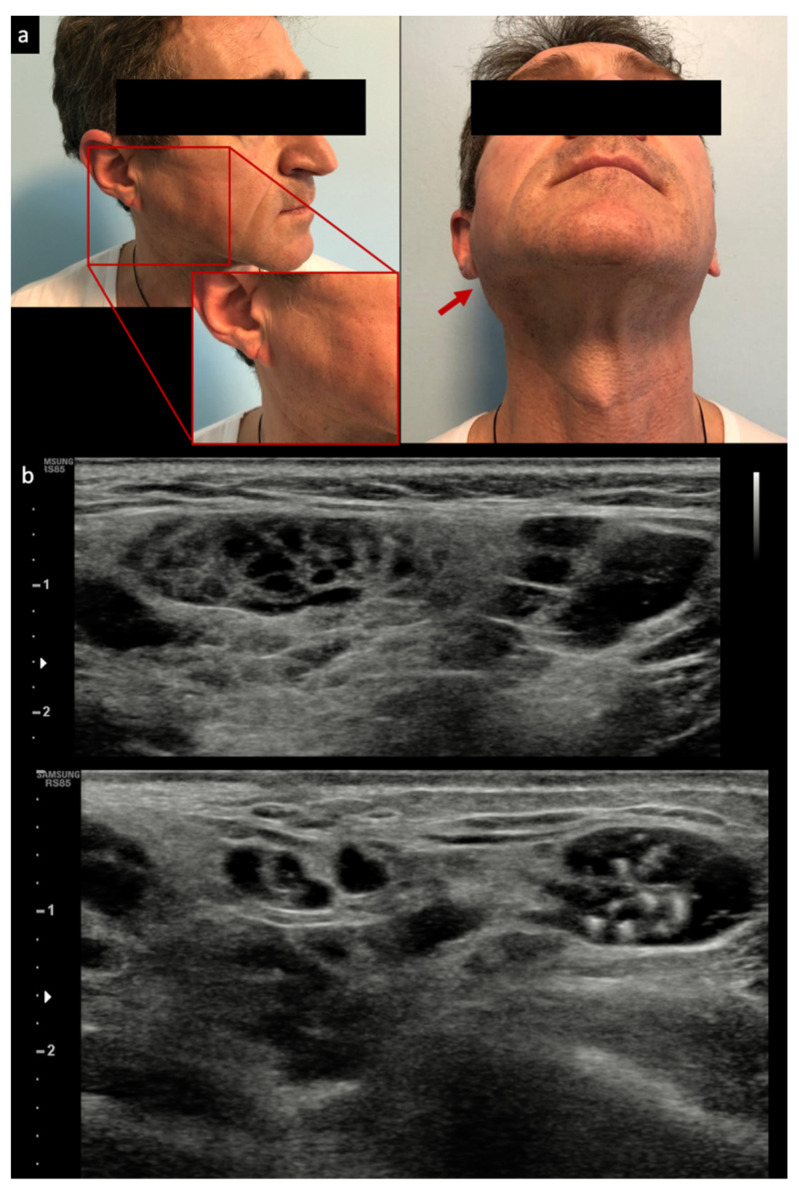
Examples of clinical parotid swelling (**a**) and appearance of mucosa–associated lymphoid tissue (MALT) lymphoma by ultrasound (**b**). Red arrow: Parotid swelling visible after cutaneous inspection.

**Figure 2 jcm-10-01171-f002:**
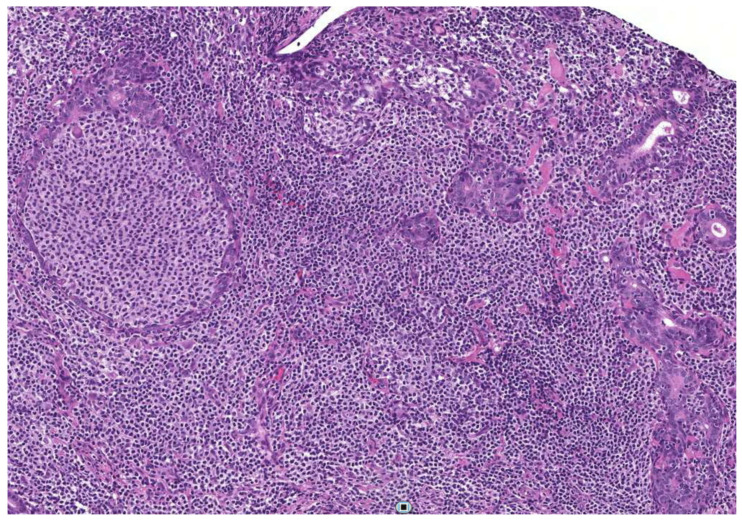
Histological picture showing a marginal zone lymphoma of the MALT type in a parotid gland characterized by diffuse coalescence of centrocyte-like lymphocytes with lymphoepithelial lesions (H&E, 200×).

**Table 1 jcm-10-01171-t001:** Clinical, laboratory, histological and imaging features to evaluate when assessing lymphoproliferative risk in pSS.

Lymphoproliferative Disorders in pSS: When to Suspect and Explore?	
Clinic signs/symptoms	- SG enlargement - Skin vasculitis - Peripheral neuropathy - Lymphadenopathy - Splenomegaly
Laboratory	- SSA/SSB positivity - Rheumatoid Factor positivity - Cryoglobulinemia - Low C4 levels - CD4+ T cells Lymphopenia - Monoclonal Component
Histology	- Focus Score ≥ 3 - presence of ectopic GC-like structures and LELs
Imaging	- US: focal, hypoechoic, dishomogeneous, with hypervascular areas with posterior acoustic enhancement, with variable morphology and margins, and with internal septa in half of the cases - PET/TC: SUVmax ≥4.7 in the parotid glands

GC: germinal center; LELs; lymphoepithelial lesion; SG: salivary gland; SUV: standardized uptake value.

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
