# Peer review of "Ultrasound and Bioptic Investigation of Patients with Primary Sjögren’s Syndrome"

_jcm, 2021, doi:10.3390/jcm10061171_

Round 1
Reviewer 1 Report
This is a well written review on imaging of the saliva glands in pSS. I think it of general interest. I think the authors could include nuclear med studies, as Tc99 and PSMA imaging measures acinar parenchyma - esp for monitoring clinical trials.
Reviewer 2 Report
Manfrè et al review the benefit of imaging and histopathological analysis of salivary glands in order to pick up lymphoproliferative complications of primary Sjögren's. This is an interesting review written by authorities in the field and could be contributory to the literature but in my opinion as it now stands it lacks supporting data for the viewpoint of the authors.
Major:
- please include a table summarizing the imaging modalities, systemic manifestations (sedimentation rate, leukocytosis e.g.) and histopathology classifications that can be used in detecting lymphoproliferation in primary Sjögren's
- a search strategy should be included even though this is a narrative review
- the results of the reliability exercise study OMERACT should be reported in the text on top of being cited
- citing a review in order to report that histology is superior to fine needle aspiration cytology does not seem accurate: please cite RCT's if available
- is there any association between peripheral blood analysis (sedimentation rate, leukocytosis, presence of autoantibodes) and imaging and/or histopathology? Please include this in the review.
Reviewer 3 Report
This review of Manfre et al is a nice summary about the present literature of ultrasound and biopsy in PSS patients.
I have some comments about this manuscript:
A focus score >1 is missing as predictor for the development of NHL in introduction section.
The aim of this review remains unclear and is not stated.
Why was MRI described in the first section? The method should be described more in detail and mentioned in the title or should be removed. MRI is not mentioned in the abstract, too.
US-guided CNB is described as safe alternative (row 183) - It remains unclear which kind of alternative is meant?
The investigation of random glands is described as major limitation. (row 287-288) Are there any references for this statement? Please comment more in detail about this limitation.
The conclusion is described for SGUS only. There are comments about biopsy (SG and PG) missing- for further research and clinical practice.
Round 2
Reviewer 2 Report
The auhors have improved the manuscript.